# Spatial Distribution of Heavy Metals in Near-Shore Marine Sediments of the Jeddah, Saudi Arabia Region: Enrichment and Associated Risk Indices

Riyadh F. Halawani [1] , Myra E. Wilson [2], Kenneth M. Hamilton [2], Fahed A. Aloufi [1], Md. Abu Taleb [1] ,
Aaid G. Al-Zubieri [3] and Andrew N. Quicksall [2],*

[1]  Department of Environmental Science, Faculty of Meteorology, Environment and Arid Land Agriculture,
    King Abdulaziz University, P.O. Box 80207, Jeddah 21589, Saudi Arabia; rhalawani@kau.edu.sa (R.F.H.);
    faloufi@kau.edu.sa (F.A.A.); taleb@manarat.ac.bd (M.A.T.)
[2]  Civil and Environmental Engineering, Southern Methodist University, Dallas, TX 75275, USA;
    wilson.m.elizabeth@gmail.com (M.E.W.); khamilton@mail.smu.edu (K.M.H.)
[3]  Marine Geology Department, Faculty of Marine Scienc, King Abdulaziz University, P.O. Box 80207,
    Jeddah 21589, Saudi Arabia; aahmed0217@stu.kau.edu.sa
*  Correspondence: aquicksall@lyle.smu.edu

**Abstract:** Red Sea coastal development has rapidly accelerated in recent decades that has led to a rise in the anthropogenic heavy metal levels in sediments. A total of 80 surficial sediment samples were collected from the shallow waters along the eastern Red Sea coast near Jeddah, Saudi Arabia. These samples were collected from three locations, designated as North, Middle and South of Jeddah, to assess the concentrations of six heavy metals: chromium (Cr), manganese (Mn), nickel (Ni), copper (Cu), zinc (Zn), and lead (Pb). The results showed that the concentrations (mg/kg) of these metals in the studied sediments follow this order: Pb (77.34 ± 150.59) > Mn (36.52 ± 37.72) > Zn (18.02 ± 23.94) > Cr (9.56 ± 5.81) > Cu (9.18 ± 13.67) > Ni (3.68 ± 4.54). The majority of the polluted sediments were recorded in the Middle and South locations. Pollution and enrichment indices such as Geo-accumulation indices (Igeo), Enrichment Factors ($E_f$), Contamination Factors ($C_f$), Pollution Load Indices (PLI), Potential Ecological Risk Indices (PERI) and Potential Toxicity Response Indices (RI) were calculated from the measured metals to establish baselines for the region and assess specific metal enrichments by location along the Jeddah coastline. The Igeo values showed that 30% of the Southern location stations are considered moderately to highly polluted. The Ef for all the studied sediments followed this order: Pb (extremely severe enrichment) > Zn > Cu > Cd > Cr (severe enrichment) > Ni (moderate enrichment).

**Keywords:** Jeddah; Red Sea; heavy metals; sediments; risk indices

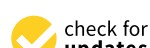



## 1. Introduction

The Jeddah coast is a part of the Red Sea, which is a semi-enclosed warm water body separating Asia from Africa and surrounded by nine countries [1]. Historically, the Red Sea has been used as the main route for merchant caravans from Europe to India and Eastern Asia and is still a major transportation routes, joining several of the world's major ocean shipping routes. Despite such heavy transit, it is relatively unpolluted, apart from localized areas [2]. With increasing coastal development, associated pollution is a rising concern for the region. Sediments can directly interact with other environmental media across numerous spatial and temporal scales [3]. Coastal sediments of Jeddah are, therefore, attractive for scholars to study heavy metal enrichment due to the sensitivity of these sediments to industrial and anthropogenic activities [4,5].

Heavy metal sourcing to the local marine environment could be anthropogenic or natural, including pollution by oil spills, wastewater discharge, effluents of desalination plants, construction activities, and marine traffic [5–8]. According to Skaldina et al., [9]



industry, transport, refuse burning, and power generation are considered as significant sources of heavy metal introduction into different environmental media. Direct or indirect wastewater discharge with its heavy metal load, leads to an increase in environmental contaminants in the associated marine environment [10,11]. Prior work showed that anthropogenic activities clearly affect nearby sediments according to the rapid industrial transformation along the coastal area of Saudi Arabia [12,13].

The concentrations of heavy metals in sediments are vital indicators to identify and classify the degree of pollution for any area of concern [14]. The concentrations of heavy metals in seawater are governed by suspended particulate matter and, ultimately, sediments [4,12]. Typically, suspended metals bind to particulate matter once discharged into aquatic systems; they settle and are sorbed by sediments [15]. While this is a complex series of processes, as shown through heavy metal sediment concentrations' correlation with pH, conductivity, salinity, and the availability of organic matter [16], sediment concentrations are linked to aquatic concentrations. Both aqueous and sediment metals can be bioavailable. Some heavy metals such as lead, cadmium, mercury, chromium, and arsenic are identified to be toxic for living organisms, whereas copper, manganese, sodium, iron, and zinc are essential metals, but can be toxic if they exceed permissible levels [17,18].

A number of studies have investigated the surface sediments of the Red Sea and showed variable heavy metal contamination. Badr et al. [8], studied heavy metals in Red Sea sediments near Jeddah and reported that Jeddah has the most contaminated sediments in comparison with the other industrial areas along the Red Sea coast. The highest concentrations of Cu, Zn, Cd, Ni, Cr, and Pb were recorded in the Al-Arbaeen and Al-Shabab inlets (central Jeddah coast), indicating their possible anthropogenic sourcing [19]. Pan et al. [5] reported that Zn, Cu, and Pb were high in sediment samples collected from near the Jeddah fish market. Moreover, Al-Mur et al. [6] reported the same conclusions for sediments near Jeddah, where the highest enrichment was found for metals such as Mn, Cr, Pb, Zn, and Cu. Bantan et al. [20] recorded high concentrations of the same heavy metals on the coastline of the Southern Corniche of Jeddah and attributed those concentrations to the influx of domestic and industrial wastewater into this zone.

The study area, including locations along the North, Middle, and South Jeddah coast (Figure 1), is highly urbanized and rich with potential anthropogenic sources of contamination, especially the industrial activities of the Industrial City located along the southern part of Jeddah coast. The Jeddah coast is connected to industrial and sewage treatment plants that may act as sources of contaminants to the region [21,22]. Therefore, a comprehensive evaluation of heavy metals contamination in sediments of the entire Jeddah shoreline needs to be covered. The specific aims of this work are to: (i) investigate of the state of heavy metal pollution along Jeddah's coast by measuring the total metal concentrations in the sediments; (ii) quantify the extent of pollution in sediments using the Geo-accumulation indices (Igeo), Enrichment Factor (EF), and Contamination Factor (CF); and (iii) determine the associated risk by using Pollution Load Index (PLI), potential ecological risk (PERI), and Potential Toxicity Response Index (RI).

Coastal areas are considered sequestration points for various pollutants generated from commercial and urban activities. Different human-induced pollutants are typically delivered to coastal sediments through atmospheric transport and fluvial processes. A higher concentration of contaminants, including heavy metals, leads to deteriorating environmental parameters, including those gauging marine ecosystem health. The state of heavy metals and their distribution in coastal sediments, therefore, needs to be identified by applying comprehensive indices for categorizing the pollution level to inform appropriate future management decisions. The data and interpretation provided here will yield valuable benchmarks, not just for the immediate area, but for the region as a whole and to similar tropical marine ecosystems worldwide.

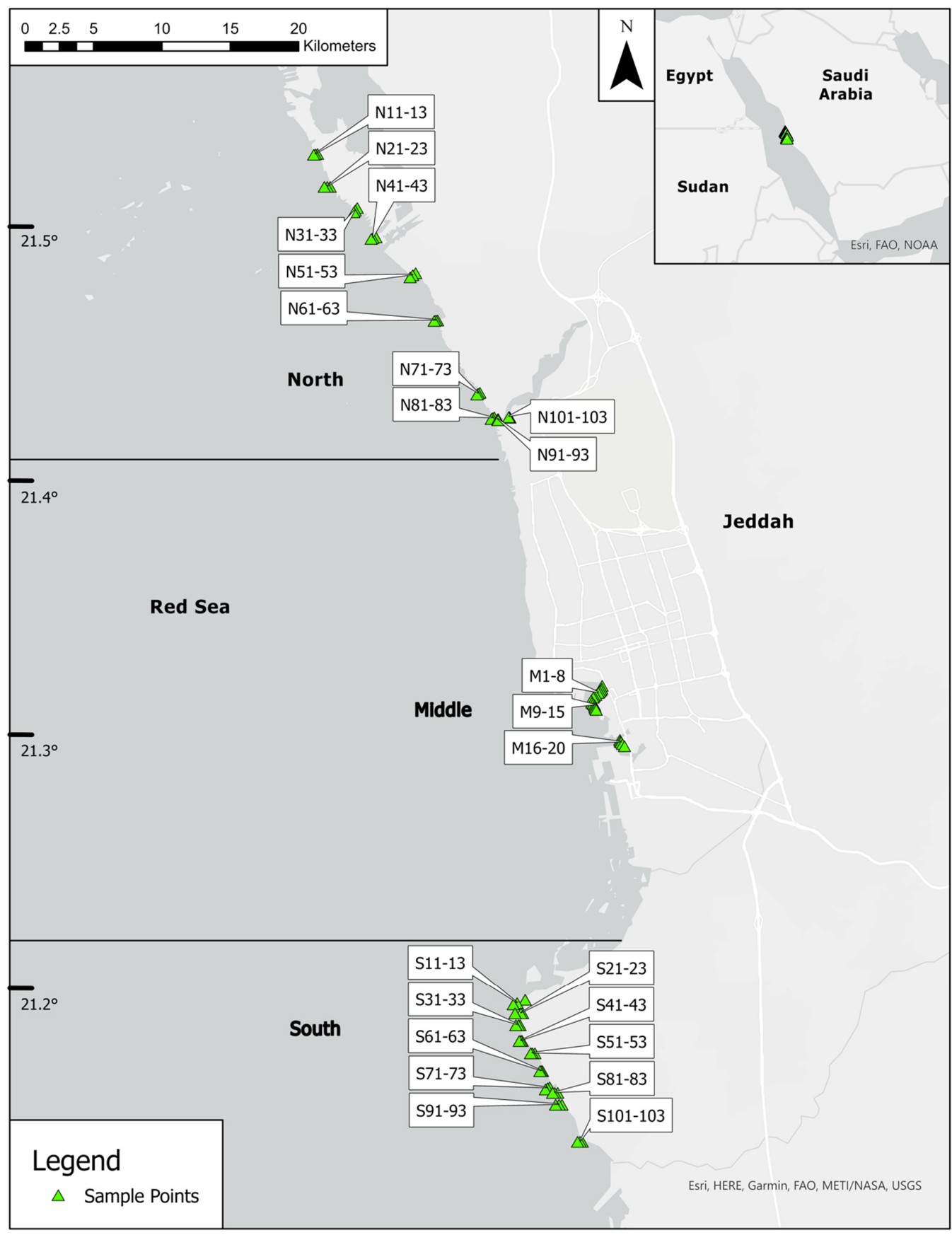

**Figure 1.** Map of the study region with broader geographical region shown in the inset. Horizontal lines delineate North, Middle, and South locations. The locations of specific sediment sampling points are labeled.

## 2. Materials and Methods

### 2.1. Sample Collection and Preparation

A total of 80 surficial sediment samples were collected from beneath shallow nearshore waters from the North, Middle, and South locations along the Jeddah coast, during the summer of 2017 (Figure 1). Specific sample location was determined using GPS; the entirety of the study area occurs between 21°52′56″ N 38° 58′12″ E and 21°14′0″ N 39°8′38″ E. The Northern and Southern samples were collected at ten stations, where at each station, 3 sample points were collected in transects perpendicular to the shoreline with distances apart of approximately 50 m. For the Middle sample points, only one sediment sample was taken for 20 stations. All the samples were collected via stainless steel soil scoop, stored in polyethylene clear flat zipper bags, and held in an icebox at 4 °C until they were delivered to the laboratory.

At the laboratory, sediment samples were placed in an oven at 105 °C until dry, ground into a powder using a mortar and pestle and sieved through a 75 μm nylon sieve. Heavy metal concentrations of the sediment samples were determined according to a modified US EPA Method 3050B [23]. 0.20 g, approximately, of the sediment powdered sample was used, as described by [8]. Sediment samples were placed in 50 mL tubes and 5 mL 5% $HNO_3$ was added to remove carbonate. Then, 20 mL of 70% $HNO_3$ was added to the samples inside a fume hood where they were digested at 85 °C on a hot block until dryness. A total of 10 mL of 5% $HNO_3$ was quantitatively added by mass after dryness to resuspend analytes.

### 2.2. Analytical Procedure and Analysis

The concentrations of heavy metals in the digested sediment samples were measured by a Thermo Scientific X-Series 2 Inductively Coupled Plasma-Mass Spectrometer (ICP-MS). Samples were run in collision cell technology with kinetic energy discrimination (CCT-KED) mode. Multi-element standards (CLMS-2A, from SPEX Certi Prep, Metuchen, NJ, USA) were prepared in a range from 0.1 to 75 ppb to establish calibration curves. To ensure quality control, replicate of blanks (5% $HNO_3$) were analyzed every 12 samples. During the sediment digestion procedure, all dilutions were made using 5% $HNO_3$ in 18.2 MΩ distilled, deionized water.

### 2.3. Assessing Pollution Impact

A number of factors and indices were calculated and interpreted to evaluate the contamination risk assessment, metal contamination levels, and potential toxicity in the study area.

#### 2.3.1. Contamination Factor ($C_f$)

Contamination factor is one of the simplest and the most common methods for determining contamination of the heavy metals by expressing the ratio between the metal concentration in the sample of the area being studied and the concentration of the same metal in a background value. The $C_f$ values where calculated using the commonly applied equation [24]:

$$C_f^i = C_s^i / C_b^i \tag{1}$$

where $C_f^i$ is the contamination factor for a specific metal, $C_s^i$ is the concentration of the specific metal in a sediment sample and $C_b^i$ is the value of the given heavy metals in a natural background [25]. Here, we employ the same background values as Badr et al. [8] used for a similar region. In this factor, if $C_f^i < 1$ refers to low contamination, $1 \leq C_f^i < 3$ refers to moderate contamination, $3 \leq C_f^i < 6$ refers to considerable contamination, and $C_f^i \geq 6$ refers to very high contamination [24].

#### 2.3.2. Enrichment Factor ($E_f$)

Different authors have measured the $E_f$ values by using several normalizing elements, such as aluminum, rubidium, iron, lithium, and scandium [4,6,16,25–28]. In this study, the

$E_f$ was calculated by using rubidium as the reference element. The $E_f$ is calculated using the following equation:

$$E_f^i = (C_i/C_{Rb})_S / (C_i/C_{Rb})_B \tag{2}$$

where $E_f^i$ is an enrichment factor for specific metal, $(C_i/C_{Rb})_S$ is the metal concentration in mg/kg in relation to rubidium in the studied sediment samples and, $(C_i/C_{Rb})_B$ are the average background shale values that are adopted by Turekian and Wedepohl, [29]. Enrichment factors values and associated sediment quality were clarified as proposed by Amin et al. [4]. Where an $E_f$ value of <1 indicates no enrichment, an $E_f$ value of <3 indicates minor enrichment, an $E_f$ value of 3–5 indicates moderate enrichment, an $E_f$ value of 5–10 indicates moderately severe enrichment, an $E_f$ value of 10–25 is severe enrichment, and an $E_f$ value of 25–50 indicates extremely severe enrichment.

### 2.3.3. Pollution Load Index (PLI)

Pollution load index is used to contrast the total concentration of elements at different sampling locations. The PLI was calculated according to the equation used by Tomlinson et al. [30].

$$PLI = \left(C_f^1 \times C_f^2 \times \ldots \times C_f^n\right)^{\frac{1}{n}} \tag{3}$$

where $C_f$ is the contamination factor and n is the number of metals.

### 2.3.4. Potential Ecological Risk Index (PERI)

PERI is one of the most common methods for determining the environmental sensitiveness to heavy metal pollution in a specific location. It combines total concentrations and toxicities to yield a weighted metric per metal. In this study, the PERI equation of Hakanson [24] was used as follows:

$$PERI = C_f^i \times T_f^i \tag{4}$$

where PERI is potential ecological risk index, $C_f^i$ is the contamination factor, and $T_f^i$ is the coefficient for the toxicity of a single metal. The corresponding $T_f^i$ listed values are Zn = Mn = 1, Cr = 2 and Cu = Ni = Pb = 5 [24]. The categories of PERI expand from low to very high potential ecological risk, according to the values of PERI, where a PERI value less than 40 indicates low risk, a PERI value between 40–80 indicates moderate risk, a PERI value between 80–160 indicates considerable risk, a PERI value between 160–320 indicates high risk, and a PERI value more than 320 indicates very high risk [24].

### 2.3.5. Potential Toxicity Response index (RI)

The RI is the total of all metals of interest that were used in the PERI. The calculated RI value followed Hakanson [24].

$$RI = \sum PERI \tag{5}$$

where RI is the sum of the PERI values for every metal of interest. The classification categories used for RI in this study followed Hakanson [24], where a RI value less than 150 indicates low risk, a RI value between 150–300 indicates moderate risk, a RI value between 300–600 indicates considerable risk, and when a RI value more than 600 indicates very high risk.

### 2.3.6. The Geo-Accumulation Index (Igeo)

The Igeo was used to define and categorize the pollution status of sediments by comparing the sediment metal concentrations with the natural background concentrations of the same metal. It was first described and calculated by Müller [31]:

$$Igeo = \ln(C_n/1.5B_n) \tag{6}$$

where, $C_n$ is the measured concentration of elements in the sediments sample, $B_n$ is the geochemical background value of the element [26], and 1.5 is the correction factor for lithogenic effects. Müller [32] proposed seven descriptive classes of Igeo: 0 indicates the unpolluted class, 1 indicates the unpolluted to moderately polluted class, 2 indicates the moderately polluted class, 3 indicates the moderately to strongly polluted class, 4 indicates the strongly polluted class, 5 indicates the strongly to very strongly polluted class, and 6 indicates the very strongly polluted class. The classes map directly to Igeo values with class 0 inclusive of values <0 and class 6 inclusive of values >6.

### 2.3.7. Multivariate Statistical Methods

Multivariate statistical techniques were applied to the metals concentration data from the study region. All location datasets, North, Middle, and South, were analyzed with multidimensional scaling (MDS) and principal component analysis (PCA) using SPSS software v20. MDS is a technique that allows mapping the similarity or common space between variables into lower-dimensional space [33]. The lower-dimensional and the original spaces are called the S-Stress which ranges from 0 to 1. Near zero indicates perfect fit and close to 1 indicates poor fit. PCA was applied to represent the relationships between the investigated metals over the study area with a small number of underlying factors that explain the pattern of correlation within a set of variables [34].

## 3. Results

### 3.1. Heavy Metals Concentrations

The average heavy metals concentrations (mg/kg) in the surface sediments of the study area were Pb (77.34 ± 150.59) > Mn (36.52 ± 37.72) > Zn (18.02 ± 23.94) > Cr (9.56 ± 5.81) > Cu (9.18 ± 13.67) > Ni (3.68 ± 4.54) see Table 1.

**Table 1.** Average concentrations of heavy metals average and standard deviations from the study area compared to other studies from the broader region. All concentrations are expressed in mg/kg. Other studies are represented by ranges or means (M) depending on data reported in the given study.

| Location | Cr | Mn | Ni | Cu | Zn | Pb | Refs. |
|---|---|---|---|---|---|---|---|
| Red Sea, Jeddah North | 8.12 ± 2.04 | 31.08 ± 19.11 | 2.74 ± 2.15 | 2.44 ± 2.23 | 11.02 ± 17.92 | 4.72 ± 2.17 | |
| Red Sea, Jeddah Middle | 12.08 ± 8.46 | 76.94 ± 51.29 | 8.14 ± 6.58 | 21.92 ± 14.18 | 41.04 ± 30.02 | 80.44 ± 99.25 | This study |
| Red Sea, Jeddah South | 9.32 ± 5.62 | 15.01 ± 9.67 | 1.59 ± 1.57 | 7.36 ± 14.25 | 8.73 ± 8.80 | 147.91± 208.91 | |
| Averages | 9.56 | 36.52 | 3.68 | 9.18 | 18.02 | 77.34 | |
| Red Sea, Jeddah, Saudi Arabia 2011 | - | - | - | 0.45–82.99 | 5.3–179 | 0.46–69.38 | [5] |
| Red Sea, Jeddah, Saudi Arabia 2017 | M = 245.96 | M = 478.45 | - | M = 251.82 | M = 623.09 | M = 362.75 | [6] |
| Red Sea, Jeddah, Saudi Arabia 2009 | 12.98–22.81 | 33.71–205.06 | 67.78–85.50 | 17.47–23.77 | 52.74–76.36 | 80.30–98.77 | [8] |
| Red Sea, North side, Saudi Arabia 2019 | M = 14.6 | - | M = 15.92 | M = 15.51 | M = 38.77 | - | [12] |
| Red Sea, Egypt 2018 | M = 53.84 | M = 291.94 | M = 15.37 | M = 7.70 | M = 27.55 | M = 4.68 | [28] |

**Table 1.** *Cont.*

| Location | Cr | Mn | Ni | Cu | Zn | Pb | Refs. |
|---|---|---|---|---|---|---|---|
| Red Sea, Jeddah, Saudi Arabia 2019 | - | M = 784 | M = 141 | M = 67 | M = 390 | M = 157 | [35] |
| Red Sea, Shuaiba, Saudi Arabia 2013 | 8.75 | 61.33 | - | 4.13 | 2.76 | 0.53 | [36] |
| Red Sea, Yanbu coast, Saudi Arabia 2018 | 4.8–201 | - | 1.37–94 | 1.35–73 | 5.88–241 | 0.08–23 | [37] |
| Red Sea, Jizan, Saudi Arabia 2017 | M = 5.64 | M = 9.58 | M = 14.32 | M = 16.39 | M = 24.74 | M = 3.86 | [38] |
| Red Sea, Saudi Arabia 2017 | M = 24.79 | M = 213.78 | M =14.23 | M = 9.33 | M = 26.79 | M = 5.55 | [39] |
| Average Chemical Composition in sediments 2001 | 74 | 680 | 40 | 40 | 65 | 17 | [40] |

These results suggest that the average Pb concentration in the sediments of the Jeddah coast (77.34 mg/kg) is consistent with prior studies near Jeddah (Table 1), falling between those found by Pan et al. [5] (69.38 mg/kg) and those of Badr et al. [8] (98.77 mg/kg). The Pb concentration in the present study is higher than that found in Yanbu sediments (an industrial coastal city north of Jeddah,) and Jizan sediments (an urban coastal city located south of Jeddah), 23.33 and 3.86 mg/kg, respectively [37,38]. Moreover, the Pb concentrations found along the Jeddah coastline were higher than those of the northern Red Sea of Egypt (4.68 mg/kg) [28].

The average Mn concentration in the present study was 36.52 mg/kg, which is lower than those for sites near Jeddah (205.06 mg/kg) and near Shuaiba (an urban coastal city located south of Jeddah) (61.33 mg/kg) [8,36]. In addition, it was lower than those of the Red Sea sediment samples that were studied by Ruiz-Compean, et al. [39] (213.78 mg/kg) from the Saudi Arabian coast and by Badawy, et al. [28] (291.94 mg/kg) for the Egyptian coast.

The Zn concentration from the present study (18.02 mg/kg) was lower than those of Jizan sediments (24.74 mg/kg) [38] and the northern Red Sea (38.77 mg/kg) [12]. On the other hand, it was higher than those from Shuaiba (2.76 mg/kg) [36].

The copper concentration showed an average of 9.18 mg/kg, which is lower than those reported for Jeddah by Pan et al. [5] and Badr et al. [8] (82.99 and 23.77 mg/kg, respectively), Yanbu by Alharbi et al. [37] (73 mg/kg), Jizan by Mortuza and Al-Misned [38] (16.39 mg/kg), northern Red Sea by Karuppasamy et al. [12] (15.51 mg/kg), and the eastern side of the Red Sea by Ruiz-Compean et al. [39] (9.33 mg/kg) (Table 1). However, copper concentrations in the present study were higher than those reported previously for sites at Shuaiba and Egypt. (4.13 and 7.70 mg/kg) [28,36].

The average chromium concentration in the present study (9.56 mg/kg) was higher than the average concentration reported by Abohassan [36] in Shuaiba (8.75 mg/kg) and Mortuza and Al-Misned [38] in Jizan (5.64 mg/kg). However, it was lower than the value that was recorded in Jeddah (22.81 mg/kg) by Badr et al. [8], in the eastern side of the Red Sea (24.79 mg/kg) by Ruiz-Compean et al. [39], in the northern Red Sea (14.6 mg/kg) by Karuppasamy et al. [12], and in the Egyptian Red Sea (53.84 mg/kg) by Badawy et al. [28].

The average concentration of nickel in the sediments of Jeddah from this study (3.68 mg/kg) was lower than those reported in Jeddah (85.50 mg/kg) by Badr et al. [8] in Jeddah, in Jizan (14.32 mg/kg) by Mortuza and Al-Misned [38], in the northern of Red Sea

(15.92 mg/kg) by Karuppasamy et al. [12], and in eastern side of the Red Sea (14.24 mg/kg) by Ruiz-Compean et al. [39].

### 3.2. Statistical Analysis

Many of the metals show a strong bivariate correlation among each other for all locations, especially between Cr-Mn and Cr-Ni (Table 2). Zn was one of the highest metals in the Northern location of the Jeddah coast, but it did not show a significant correlation with other elements suggesting a different sourcing of Zn in the North relative to other metals. In the Middle location, Zn had a strong significant positive correlation with Cr (r = 0.83), Mn (0.94), Ni (r = 0.88) and Cu (0.96), indicating that these heavy metal pollutants were most likely derived from the same source of contamination. In the South location of the Jeddah coast, Cu was unique in that is did not express even weak correlations with any other analyte. In the same area, Pb showed weak positive correlations with Cr, Mn, and Zn (r = 0.57, 0.62, and 0.65, respectively).

**Table 2.** Correlation coefficients for heavy metals concentrations in marine sediment samples from each location near Jeddah, Pollution Load Index (PLI) and the Potential Toxicity Response Index (RI).

| North | Cr | Mn | Ni | Cu | Zn | Pb | PLI | RI |
|---|---|---|---|---|---|---|---|---|
| Cr | 1.00 | | | | | | | |
| Mn | 0.72 | 1.00 | | | | | | |
| Ni | 0.82 | 0.87 | 1.00 | | | | | |
| Cu | 0.72 | 0.82 | 0.92 | 1.00 | | | | |
| Zn | 0.28 | −0.03 | 0.13 | 0.13 | 1.00 | | | |
| Pb | 0.28 | 0.22 | 0.24 | 0.25 | −0.20 | 1.00 | | |
| PLI | 0.82 | 0.87 | 0.96 | 0.94 | 0.19 | 0.38 | 1.00 | |
| RI | 0.72 | 0.63 | 0.74 | 0.75 | 0.18 | 0.77 | 0.85 | 1.00 |
| Middle | | | | | | | | |
| Cr | 1.00 | | | | | | | |
| Mn | 0.95 | 1.00 | | | | | | |
| Ni | 0.98 | 0.97 | 1.00 | | | | | |
| Cu | 0.82 | 0.94 | 0.88 | 1.00 | | | | |
| Zn | 0.83 | 0.94 | 0.88 | 0.96 | 1.00 | | | |
| Pb | 0.51 | 0.42 | 0.48 | 0.39 | 0.25 | 1.00 | | |
| PLI | 0.96 | 0.94 | 0.96 | 0.87 | 0.83 | 0.67 | 1.00 | |
| RI | 0.57 | 0.50 | 0.55 | 0.47 | 0.33 | 1.00 | 0.73 | 1.00 |
| South | | | | | | | | |
| Cr | 1.00 | | | | | | | |
| Mn | 0.85 | 1.00 | | | | | | |
| Ni | 0.83 | 0.94 | 1.00 | | | | | |
| Cu | 0.09 | 0.21 | 0.25 | 1.00 | | | | |
| Zn | 0.92 | 0.87 | 0.89 | 0.18 | 1.00 | | | |
| Pb | 0.57 | 0.62 | 0.65 | 0.17 | 0.46 | 1.00 | | |
| PLI | 0.86 | 0.93 | 0.95 | 0.22 | 0.88 | 0.76 | 1.00 | |
| RI | 0.57 | 0.63 | 0.66 | 0.20 | 0.47 | 1.00 | 0.77 | 1.00 |

To further explore covariance and, therefore, potential metal co-sourcing, multivariant techniques were applied as described in the methods. The output of factor analysis revealed

that ~84.40% of data variance can be explained by the first two principal components (Table 3). Combined interpretation of the PCA, MDS, and bivariant correlation data will be discussed below.

**Table 3.** Principal Component Analysis of the heavy metals from the study area.

| Component | Initial Eigenvalues | | | Extraction Sums of Squared Loadings | | | Rotation Sums of Squared Loadings | | |
|---|---|---|---|---|---|---|---|---|---|
| | Total | % of Variance | Cumulative % | Total | % of Variance | Cumulative % | Total | % of Variance | Cumulative % |
| 1 | 4.024 | 67.063 | 67.063 | 4.024 | 67.063 | 67.063 | 3.827 | 63.785 | 63.785 |
| 2 | 1.040 | 17.341 | 84.404 | 1.040 | 17.341 | 84.404 | 1.237 | 20.619 | 84.404 |
| 3 | 0.491 | 8.179 | 92.583 | - | - | - | - | - | - |
| 4 | 0.256 | 4.269 | 96.853 | - | - | - | - | - | - |
| 5 | 0.162 | 2.695 | 99.548 | - | - | - | - | - | - |
| 6 | 0.027 | 0.452 | 100.000 | - | - | - | - | - | - |

*3.3. Risk Assessments*

Holistic risk assessment and degree of pollution for the study region was done by calculating a series of factors and indices from sediment metal concentrations including: Contamination Factors ($C_f$); Enrichment Factor ($E_f$); Pollution load index (PLI); Potential Ecological Risk Index (PERI); Potential Toxicity Response index (RI); and Geo-accumulation index (Igeo).

The $C_f$ of the recorded heavy metals showed that the sediments of Jeddah coast have, in general, a low degree of contamination. Pb is an exception, which showed a considerable contamination (4.02) and very high contamination (7.39) in the Middle and South locations, respectively (Table 4). Zhuang and Gao [41] interpreted any $E_f$ values for a metal > 1.5 to be anthropogenic in source. In general, the order of enrichment of the surficial sediments of the Jeddah coast was Pb (extremely severe enrichment) > Mn > Cu > Zn > Cr (severe enrichment) > Ni (moderate enrichment). The $E_f$ of Pb, Cu, and Cr were the highest in the southern stations (2397.99, 46.60, 42.79, respectively), while the $E_f$ of Pb, Zn and Cr were also high in the North location (71.50, 36.73, and 26.80) (Table 5).

**Table 4.** Contamination Factor and Potential Ecological Risk Index for metals of interest from different locations in the Jeddah region.

| Elements | Location | Contamination Factor | | | Potential Ecological Risk Index | | |
|---|---|---|---|---|---|---|---|
| | | Average ± SD | Maximum | Minimum | Average ± SD | Maximum | Minimum |
| Cr | N | 0.09 ± 0.02 | 0.138 | 0.055 | 0.180 ± 0.05 | 0.28 | 0.11 |
| | M | 0.13 ± 0.09 | 0.361 | 0.029 | 0.27 ± 0.19 | 0.72 | 0.06 |
| | S | 0.10 ± 0.06 | 0.226 | 0.027 | 0.21 ± 0.12 | 0.45 | 0.05 |
| Mn | N | 0.04 ± 0.02 | 0.090 | 0.007 | 0.04 ± 0.02 | 0.09 | 0.007 |
| | M | 0.09± 0.06 | 0.220 | 0.010 | 0.09 ± 0.060 | 0.22 | 0.01 |
| | S | 0.02 ± 0.01 | 0.050 | 0.004 | 0.018 ± 0.011 | 0.05 | 0.004 |
| Ni | N | 0.04 ± 0.03 | 0.145 | 0.008 | 0.20 ± 0.16 | 0.72 | 0.04 |
| | M | 0.11 ± 0.10 | 0.352 | 0.009 | 0.59 ± 0.48 | 1.76 | 0.05 |
| | S | 0.02 ± 0.02 | 0.088 | 0.001 | 0.11 ± 0.11 | 0.44 | 0.01 |
| Cu | N | 0.054 ± 0.05 | 0.250 | 0.007 | 0.27 ± 0.25 | 1.25 | 0.03 |
| | M | 0.48 ± 0.32 | 1.326 | 0.068 | 2.43 ± 1.58 | 6.63 | 0.34 |
| | S | 0.16 ± 0.31 | 1.700 | 0.005 | 0.82 ± 1.58 | 8.49 | 0.03 |
| Zn | N | 0.12 ± 0.19 | 1.033 | 0.007 | 0.12 ± 0.19 | 1.03 | 0.007 |
| | M | 0.43 ± 0.32 | 1.338 | 0.053 | 0.43 ± 0.32 | 1.34 | 0.05 |
| | S | 0.092 ± 0.093 | 0.401 | 0.001 | 0.09 ± 0.093 | 0.40 | 0.001 |

**Table 4.** *Cont.*

| Elements | Location | Contamination Factor | | | Potential Ecological Risk Index | | |
|---|---|---|---|---|---|---|---|
| | | Average ± SD | Maximum | Minimum | Average ± SD | Maximum | Minimum |
| Pb | N | 0.23 ± 0.11 | 0.592 | 0.084 | 1.180 ± 0.54 | 2.96 | 0.42 |
| | M | 4.02 ± 4.96 | 18.636 | 0.085 | 20.11 ± 24.81 | 93.18 | 0.42 |
| | S | 7.395 ± 10 45 | 44.157 | 0.072 | 36.97 ± 52.23 | 220.78 | 0.36 |

**Table 5.** The Enrichment Factor and the Geo-accumulation index of the sediment samples from different locations.

| Elements | Location | Enrichment Factor Normalized by Rb | | | The Geo-Accumulation Index | | |
|---|---|---|---|---|---|---|---|
| | | Average ± SD | Maximum | Minimum | Average ± SD | Maximum | Minimum |
| Cr | N | 26.8 ± 14.73 | 66.43 | 8.21 | −2.48 ± 0.25 | −2.39 | −3.31 |
| | M | 6.97 ± 2.68 | 11.17 | 2.64 | −2.64 ± 0.69 | −1.42 | −3.96 |
| | S | 42.79 ± 25.68 | 103.93 | 8.78 | −2.87 ± 0.64 | −1.89 | −4.01 |
| Mn | N | 8.95 ± 4.27 | 20.63 | 3.78 | −3.91 ± 0.64 | −2.78 | −5.37 |
| | M | 4.60 ± 1.90 | 9.18 | 1.54 | −3.06 ± 0.77 | −1.94 | −4.80 |
| | S | 6.76 ± 3.25 | 17.51 | 2.53 | −4.63 ± 0.60 | −3.41 | −5.90 |
| Ni | N | 8.27 ± 2.21 | 14.85 | 5.12 | −3.91 ± 0.79 | −2.34 | −5.23 |
| | M | 5.72 ± 2.50 | 10.89 | 1.12 | −2.88 ± 0.89 | −1.45 | −5.11 |
| | S | 7.23 ± 5.16 | 28.16 | 0.57 | −4.66 ± 1.05 | −2.84 | −7.34 |
| Cu | N | 10.71 ± 5.46 | 30.56 | 3.66 | −3.73 ± 0.97 | −1.79 | −5.42 |
| | M | 24.68 ± 9.32 | 44.31 | 10.24 | −1.36 ± 0.75 | −0.12 | −3.10 |
| | S | 46.60 ± 75.50 | 410.99 | 4.52 | −3.21 ± 1.37 | 0.12 | −5.63 |
| Zn | N | 36.73 ± 98.70 | 557.97 | 3.58 | −3.24 ± 1.14 | −0.37 | −5.34 |
| | M | 21.06 ± 9.01 | 42.69 | 8.45 | −1.55 ± 0.83 | −0.11 | −3.35 |
| | S | 29.76 ± 20.60 | 71.18 | 0.64 | −3.45 ± 1.39 | −1.32 | −7.58 |
| Pb | N | 71.50 ± 49.65 | 238.96 | 13.10 | −1.95 ± 0.44 | −0.93 | −2.88 |
| | M | 243.99 ± 284.44 | 946.31 | 10.18 | 0.04 ± 1.57 | 2.52 | −2.87 |
| | S | 2397.99 ± 3357.43 | 14,458.22 | 31.16 | 0.14 ± 1.97 | 3.38 | −3.04 |

The average PERI values (<40) showed a low risk for all the studied stations of the Jeddah coast (Table 4). However, the PERI values of Pb demonstrate that some stations of the Middle location of the Jeddah coast, such as stations M18 and M14, should be given special attention due to the existence of high PERI values (93.17 and 71.71, respectively) that categorize the Middle location under moderate risk (Table 4). The same was recorded for the S1, S2, S3, and S7 of Southern location where the PERI values were 144.62, 63.16, 76.68, and 46.65, respectively, suggesting occurrence of considerable risk (Table 4).

The Igeo was used to assess the concentration of metals compared to average continental crust concentrations as reference values. The average Igeo values are summarized in Table 5. The Igeo values in the studied sediments of all locations were low and categorized not polluted except the Igeo values of Pb in the Middle and Southern locations. In the Middle, 30% of the stations were categorized as unpolluted to moderately polluted and 10% of the stations were categorized as moderately polluted, whereas 20% of the South stations were classified as unpolluted to moderately polluted and 20% as moderately polluted.

The PLI of most of the studied sites was >1.0 (Table 6). PLI values were 0.06, 0.30, and 0.11 for North, Middle and South locations of the Jeddah coast, respectively. The Potential Toxicity Response index (RI) showed a value of <150 for all the studied stations, suggesting a low risk for the study area (Table 6).

**Table 6.** Pollution Load Index and the Potential Toxicity Response Index for the North, Middle and South locations in the study area.

|  | Pollution Load Index | | | Potential Toxicity Response Index | | |
|---|---|---|---|---|---|---|
|  | Average ± SD | Maximum | Minimum | Average ± SD | Maximum | Minimum |
| N | 0.067 ± 0.04 | 0.181 | 0.020 | 1.98 ± 0.80 | 3.98 | 0.71 |
| M | 0.30 ± 0.21 | 0.780 | 0.030 | 23.94 ± 25.96 | 99.38 | 1.03 |
| S | 0.11 ± 0.10 | 0.340 | 0.008 | 38.19 ± 52.72 | 223.10 | 0.45 |

## 4. Discussion

The Northern location of the Jeddah coast showed higher average concentrations of Mn and Zn (31.08 and 11.02 mg/kg, respectively) especially for samples N32, N72, N101 (Figure 1). These samples are near resorts, a boat station, workshops for general maintenance and body shops for marine transportation such as boats and jet skis. Moreover, many anthropogenic activities occur such as wastewater discharge and boat painting/repair around these stations which may cause the observed elevations in sediment metals concentrations [12,13,42,43]. Furthermore, most of the resorts and beach houses in the North location are still not connected to the municipal drainage network and discharge wastewater directly to the sea. Some authors [19,44,45] report that discharge of wastewater may lead to the rise in concentrations of Mn in surface sediments. Zinc in the Northern stations may be due to pesticide usage, antifouling paints, detergents, and dispersant improvers for lubricating oil and antioxidants [46,47]. Most of the resorts are using pesticides to control pests and weeds which has been shown elsewhere to yield elevated Zn levels. In addition, batteries and the wear of automobile tires could be substantial anthropogenic sources of zinc near marine environments [8,47]. Indeed, most of the marinas use old tire products as anti-scratch material between the body of ships and the base of the marina.

Most of the metals were higher in the Middle location of the Jeddah coast (Figure 1) than the other locations especially Pb, Mn, Zn and Cu (80.44, 76.94, 41.04, and 21.92 mg/kg, respectively) (Table 2). According to Abu-Zied and Orif [45], the marine environment in the vicinity of the northern stations of the Middle location (Stations M1 to M10) is used as a dumping zone for industrial and sewage wastewaters via sewage outflow pipes. This has led to the increase of organic matter which could act as a significant carrier phase for heavy metals and has been repeatedly shown to play a key role in the bioavailability, reactivity, and mobility of metals in sediments [48]. In the present study, the prominent high concentration of Zn in the Middle location (Stations M6–M14) could be due to the outflow of organic matter from sewage sludge, industrial discharge, and antifouling paints [19,34].

The Middle location is located near the biggest desalination plant on the Jeddah coast. This plant is a major consumer of gasoline for onsite power production. The use of gasoline may be a possible source for the Pb in local sediments especially at stations M14, M15, M16, and M18 [43] (Figure 1). Elevated Mn concentrations recorded at stations M17–M20 occur near the main seaport of Jeddah; here, there are body shops for small fishing ships, where the residual ship wastes and antifouling paints are dominant. According to Abu-Zied et al. [19], domestic sewage from the municipal drainage network of Jeddah City could be one of the reasons for elevated Mn concentrations. The high Cu concentrations in the sediments of the Middle stations may again be attributed to the availability of high amounts of organic matter (OM) in the sediments given its well-established high affinity for OM [46,47]. Additionally, a public park that was renovated and expanded between station M6 and station M15, may add a new source of OM to the location.

Pb was the most elevated heavy metal in the southern portion of the Middle location and particularly in the northern portion of the South location, reaching up to 600 mg/kg (Figure 2). Most of the industrial activities of Jeddah City are located in the south and that could be identified as a primary potential source for pollutants at the studied southern stations (Figure 2). Industrial contamination, therefore, is expected in the South location near to the industrial city. This complex is located near the northern end of the South location. As an example, a major petrol refinery is located near the northern stations of the South location. As expected from this interpretation, the most affected stations of the South location of the Jeddah coast are located in the northern part of this location. Additionally, station 7 is one of the most Pb-impacted stations due possibly to its proximity to the boat station and small local fish market.

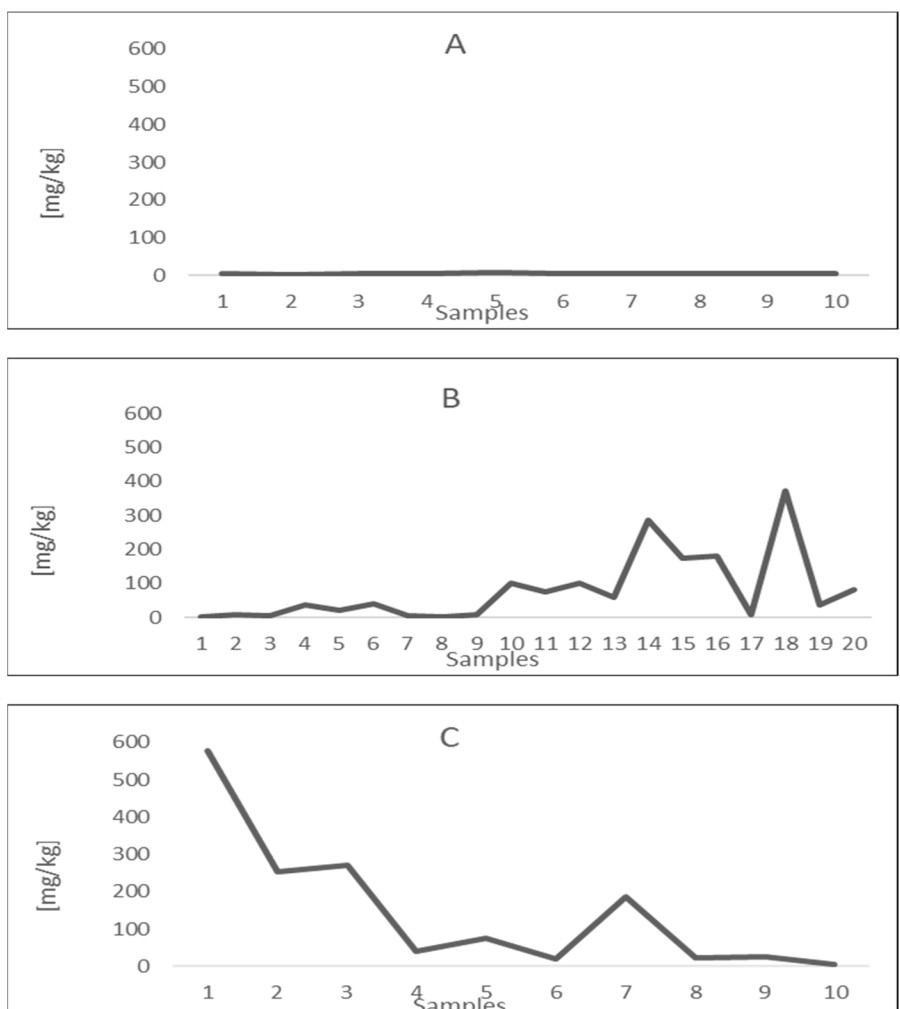

**Figure 2.** Pb concentrations in from all locations: (**A**) North, (**B**) Middle, and (**C**) South. The x-axis represents latitude in that samples progress from north to south with an increasing sample number at each location.

The contamination factor analysis showed that about 40% of the middle stations were classified as being contaminated and very highly contaminated with Pb. Multiple sources of pollution at the Middle location of the Jeddah coast such as the desalination plant, industrial and sewage wastewater outflow pipes, the seaport, the fish market, and excessive shipping activities explain the north to south trends of PLI and RI values (Figure 3). There is strong correlation between Pb concentrations and RI in the Middle location (r = 1.00). However, the correlation between Pb concentrations and PLI was the weakest of all metals (r = 0.67) at the Middle location, which may indicate that Pb is not controlling the pollution at this

location (Table 6). It is notable that both the PLI and RI trend higher as sampling progresses southward in the Middle location and both indices trend higher sampling northward in the South location (Figure 3).

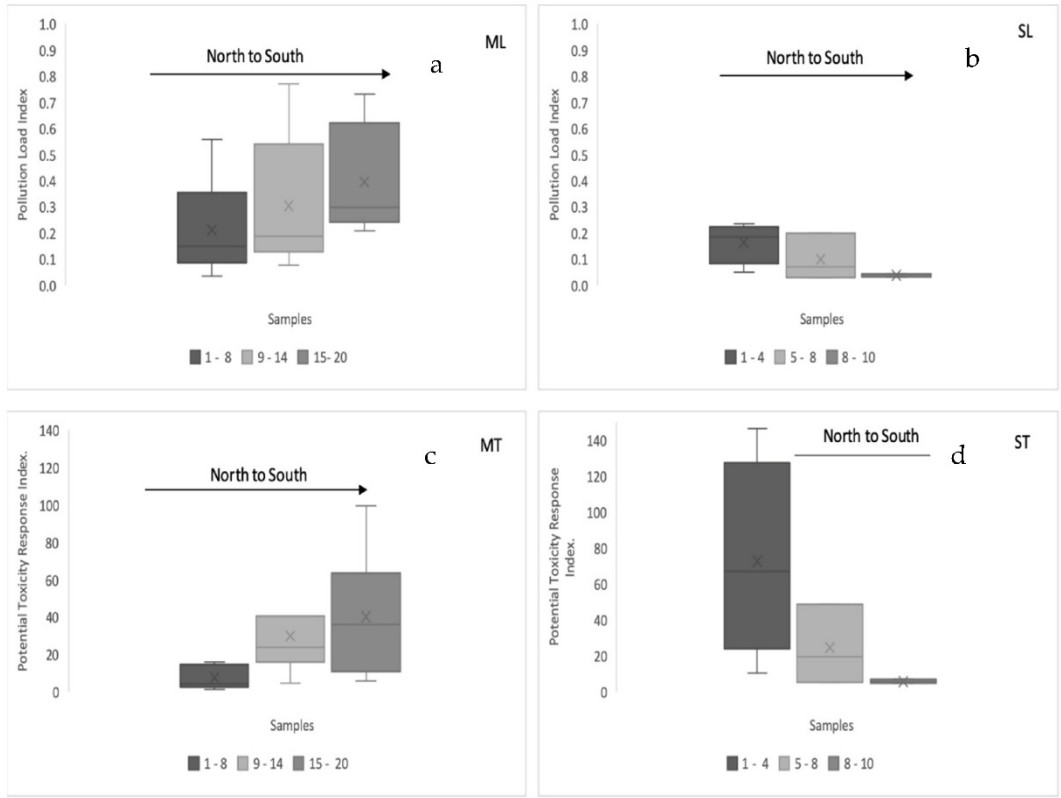

**Figure 3.** Pollution Load Index for Middle (**a**) and South (**b**) locations, and Potential Toxicity Response Index for Middle (**c**) and South locations (**d**).

Measuring metals concentrations in sediment as a function to how distal the sediment is from shore can potentially provide useful information on likelihood of sourcing. The concentrations of Cr, Mn, Ni, Cu, and Zn were high for all the nearshore stations, however, Pb concentrations were higher for the offshore, deepest stations in the northern stations as well as in the central stations of transects of the southern part of Jeddah coast (Table 7). The concentrations of Pb are known to increase offshore in the region [44]. The current study generally shows a similar trend. Cr, Mn, Ni, Cu, and Zn trends all suggest sourcing from near-shore activities. The aberration of Pb increases with distance from shore in these regions, as expected from literature, and is also likely due to near shore sourcing. About 36% of the South location stations were classified as very highly contaminated with Pb via contamination factor calculations. Further, Pb was the most enriched metal in these stations, reaching up to 88%, which may indicate that the source of Pb is anthropogenic [8]. Figure 3 shows that the PLI values for the stations S1–S4 were low; however, RI values were higher, which may indicate the occurrence of highest toxicity at these stations. The strong correlation of Pb with RI (r = 1.00) shows that the Pb is governing the pollution in the southern stations of Jeddah coast (Table 6).

To further evaluate sourcing, we turn to the statistical analysis of the coexpression of metals. The bivariate correlations discussed above and shown in Table 2 are supported by the multivariate statistical analysis. The MDS of the Northern location (Figure 4a) shows that the distribution of metals is clustered into two groups, with S-Stress 0.012 and Tucker's Coefficient of Congruence 0.99. The first group consists of Ni, Cu, Pb, and Cr. This group has a close similarity, indicating a similar source such as a siliciclastic influx (e.g., windblown dust) that accumulated along the coastline. The second includes Zn and Mn, which could

be accumulated on the sediments from another source such as wastewater discharge or boat painting/repair waste that is discharged in this zone. For the Middle location, the MDS has also clustered the concentration of metals into two groups similar to the northern part with differentiation of the second group that includes Pb (Figure 4b). This means that the lead concentration has been increased in this zone to reach a moderate contamination level, indicating to additional source for this metal (e.g., the desalination plant). However, the MDS of the Southern location shows that there is a high similarity among most of the metals in one group except Pb (Figure 4c), suggesting a high accumulation of lead to high levels of contamination. This is could be due to industrial activities along the southern coast of Jeddah. These results are also confirmed by the PCA of the overall study area, which shows that Pb is clustered far away from other metals (Figure 4d).

**Table 7.** The average concentrations (mg/kg) of heavy metals in the North and South locations with different distances from shoreline (Near—at the shoreline; Middle—25–30 m from the shoreline; Far—50–55 m from the shoreline).

|  | Cr | Mn | Ni | Cu | Zn | Pb |
|---|---|---|---|---|---|---|
|  |  |  | North |  |  |  |
| Near | 9.33 | 32.72 | 3.49 | 3.25 | 21.23 | 5.09 |
| Middle | 8.08 | 32.46 | 2.89 | 2.32 | 5.8 | 3.93 |
| Far | 6.96 | 28.08 | 1.85 | 1.75 | 6.03 | 5.14 |
|  |  |  | South |  |  |  |
| Near | 10.53 | 19.18 | 2.15 | 13.74 | 11.39 | 133.31 |
| Middle | 8.5 | 12.54 | 1.15 | 4.38 | 5.76 | 206.28 |
| Far | 8.93 | 13.3 | 1.3 | 3.23 | 6.42 | 104.13 |

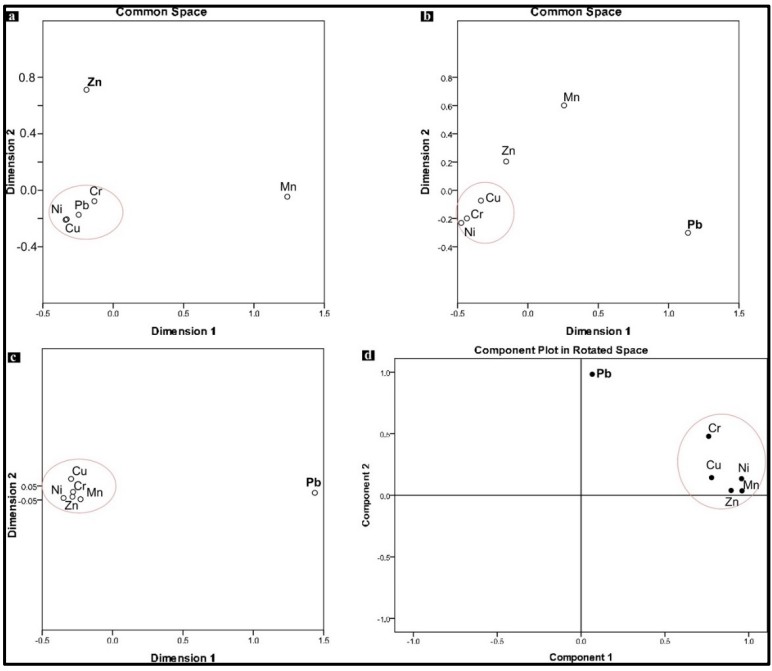

**Figure 4.** Multidimensional scaling of heavy metals in Jeddah coastline sediments, (**a**) MDS of metals concentration of the North location, (**b**) MDS of metals concentrations of the Middle location, (**c**) MDS of metals concentrations of the South location, and (**d**) PCA of heavy metals concentrations over the entire study region. Ellipses are drawn to demonstrated clusters of elements.

## 5. Conclusions

A total of 80 stations were chosen for collection of surficial marine sediments from North, Middle, and South locations of the nearshore Jeddah coast. The concentrations of six heavy metals (Cr, Mn, Ni, Cu, Zn, and Pb) in the collected sediments were determined. The average concentrations of these metals in surface sediments showed that Pb > Mn > Zn > Cr > Cu > Ni. The Enrichment Factors clearly show Pb was the most abundant heavy metal relative to background values in the study area. The Igeo values of the Pb in the Middle stations showed that 10% of the stations were categorized as moderately polluted, whereas 20% of the south locations were classified as moderately polluted. This is further supported by the contamination factor values as seen in Table 4. Furthermore, Pb was the most enriched metal in the study area, which is consistent with an anthropogenic source of Pb. The RI values in the South stations, especially the northern stations within the South location, indicate high Pb pollution in the area.

The study results suggest the potential sources of pollutants in the Middle stations of the Jeddah coast could be anthropogenically sourced from released from the desalination plant, the outflow tubes of industrial and municipal wastewaters and waste of ship maintenance. The South location of the Jeddah coast, however, is interpreted to be polluted by the industrial activity wastes from the industrial city. This study recommends that appropriate management strategies should be applied for the North location of the Jeddah coast to control potential pollution sources and prevent permanent hazards to marine ecology currently documented elsewhere. This study indicates that the Middle location is impacted by various sources due to the highest activity in the region. Stringent management practices are suggested to limited further metals contamination to the area. The Southern location of the Jeddah coast is the most toxic location; it, therefore, needs more effort and strong regulations to treat and recover the marine environment. The area needs more research towards exact sourcing of contaminants, specifically lead. A further study of metals speciation in marine sediments of Jeddah coast should be carried out to yield more information about the bioavailability and mechanism of deposition of these metals and their sourcing. Moreover, tracer studies, such as isotopic studies, paired with Pb speciation could be used to investigate the sources of the lead in sediments, water, and even in airborne aerosols near to the shoreline.

Heavy metals in near-shore marine sediments and their associated presence in marine waters are a threat to marine ecosystems. A comprehensive evaluation of heavy metals contamination in such sediments, therefore, is prudent for sensitive regions in close proximity to development. The results of this study demonstrate the use of different environmental indicators to identify current environmental health status and risk associated with anthropogenic activities. Further, bi- and mutlivariant statistical techniques suggest sourcing of coexpressed pollutants. The findings will provide a valuable set of benchmarks for future research. Additionally, they will be a highly valuable guide to environmental decision makers in KSA and counterparts in other regional countries. Finally, the techniques employed and the correlations demonstrated can further aid similar work in analogous ecosystems worldwide.

**Author Contributions:** Conceptualization, A.N.Q. and R.F.H.; methodology, A.N.Q. and R.F.H.; software, R.F.H. and K.M.H.; validation, A.N.Q. and K.M.H.; formal analysis, A.N.Q., R.F.H., M.E.W., F.A.A., M.A.T. and K.M.H.; resources, A.N.Q. and R.F.H.; data curation, A.N.Q. and R.F.H.; writing—original draft preparation, R.F.H.; writing—review and editing, A.N.Q., R.F.H., M.E.W. and K.M.H.; visualization, A.G.A.-Z., R.F.H. and M.E.W.; supervision, A.N.Q.; project administration, A.N.Q.; funding acquisition, R.F.H. All authors have read and agreed to the published version of the manuscript.

**Funding:** This research received no external funding.

**Institutional Review Board Statement:** Not applicable.

**Informed Consent Statement:** Not applicable.

**Data Availability Statement:** Not applicable.

**Acknowledgments:** The authors would like to thank King Abdulaziz University, Jeddah, Saudi Arabia and the Lyle School of Engineering, Southern Methodist University for the materials and the laboratories.

**Conflicts of Interest:** The authors declare no conflict of interest.

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
