# Peer review of "Spatial Distribution of Heavy Metals in Near-Shore Marine Sediments of the Jeddah, Saudi Arabia Region: Enrichment and Associated Risk Indices"

_jmse, doi:10.3390/jmse10050614_

Round 1

Reviewer 1 Report

This article was well edited based on original research results.

However followings should have to revised before publishing as one of research articles for international journal.

1) Generalization

The author(s) should have to explain why these results are important in the main body(Ex.: Introduction) of this research article in order to make generalization not only Saudi Arabia but also whole world.

2) Application

So what?, Where and how can we apply based on this research article's results.

3) Keywords

Heavy metals ---> Harmful heavy metals

4) Figure 1

Please add coordination system in X and Y axis(Ex.: N 00.0000, E 00.0000)

5) Table 1, 2, 3, 4, 5

Please use same effective numbers at least 3 or more  in a Table

Ex.:

Effective numbers 3: 8.12, 31.1, 0.460, etc

Effective numbers 4: 8.120, 31.10, 0.4600, etc

Please check the format of each table and remove all vertical line(s) in a Table

6) Near / Middle / Far in Table 6

Please add more detailed information such as distance from shore line

Author Response

Thank you, Reviewer 1, for taking the time to provide these valuable comments. Please find below our responses to your comments highlighting the modifications made to the final version of the paper.

  • Generalization: you commented “The author(s) should have to explain why these results are important in the main body(Ex.: Introduction) of this research article in order to make generalizations not only Saudi Arabia but also the whole world”.

The authors agree that more explicitly stating the broader impacts would make for a stronger manuscript. Therefore, the following paragraph was added to the manuscript’s introduction:

Coastal areas are considered sequestration points for various pollutants generated from commercial and urban activities. Different human-induced pollutants are typically delivered to coastal sediments through atmospheric transport and fluvial processes. A higher concentration of contaminants, including heavy metals, leads to deteriorating environmental parameters, including those gauging marine ecosystem health. The state of heavy metals and their distribution in coastal sediments, therefore, needs to be identified by applying comprehensive indices for categorizing the pollution level to inform appropriate future management decisions. The data and interpretation provided here will yield valuable benchmarks, not just for the immediate area, but for the region as a whole and to similar tropical marine ecosystems worldwide.

  • Application: you commented “So what?, Where and how can we apply this based on this research article's results”.

The authors feel this is highly related to the first comment and continued to alter the text to accommodate the request. Accordingly, the following paragraph was added to the conclusion:

Heavy metals in near-shore marine sediments and their associated presence in marine waters are a threat to marine ecosystems.  A comprehensive evaluation of heavy metals contamination in such sediments, therefore,  is prudent for sensitive regions in close proximity to development. The results of this study demonstrate the use of different environmental indicators to identify current environmental health status and risk associated with anthropogenic activities. Further, bi- and mutlivariant statistical techniques suggest sourcing of coexpressed pollutants.  The findings will provide a valuable set of benchmarks for future research. Additionally, they will be a highly valuable guide to environmental decision makers in KSA and counterparts in other regional countries. Finally, the techniques employed and the correlations demonstrated can further aid similar work in analogous ecosystems worldwide.

  • Keywords, you commented “Heavy metals ---> Harmful heavy metals”

Given the journal’s limitation on keywords and the convention of associated literature, the authors feel “heavy metals” is the appropriate keyword in this field.

  • You commented “Figure 1, Please add coordination system in X and Y axis (Ex.: N 00.0000, E 00.0000)”.

       Thank you very much for this comment; we have added it to the figure.

  • You commented “Table 1, 2, 3, 4, 5, Please use same effective numbers at least 3 or more in a Table. Ex.: Effective numbers 3: 8.12, 31.1, 0.460, etc., Effective numbers 4: 8.120, 31.10, 0.4600, etc. , Please check the format of each table and remove all vertical line(s) in a Table ”

We reviewed the numbers for all tables, the data is reported with the correct number of significant figures based on the analysis.

You commented “Near / Middle / Far in Table 6 , Please add more detailed information such as distance from shore line”.

The table’s full caption now reads: “The average concentrations (mg/kg) of heavy metals in the North and South locations with different distances from shoreline (Near: at the shoreline, Middle: 25-30 m from the shoreline, Far: 50-55 m from the shoreline). Color indication marks high (red), medium (yellow), and low (green) distance by average concentration per metal.”

Reviewer 2 Report

In the manuscript entitled “Spatial Distribution of Heavy Metals in Near-shore Marine Sediments of the Jeddah, Saudi Arabia Region: Enrichment and Associated Risk Indices” the authors were collected eighty samples, analyzed them by ICP-MS technique, and estimated their concentration in the gathered materials. Overall, the work is well done, carefully thought, and performed, and the manuscript is well written and easy to read and follow. All experimental and computational methods are well explained. The results presented are significant, robust and their presentation, interpretation, and conclusions are supported by other data present in the literature.

Other Specific comments:

Why is the letter M inserted in the Table 1? Whether it is necessary?

Another comment also applies to Table 1. The authors have provided appropriate references, but adding the year of the study (in the table under/next to literature reference) of a particular metal would give more information.

I hope that the published results will have an impact on environmental protection of the Jeddah, Saudi Arabia Region.

Author Response

First of all, thank you very much for your complimentary review we hope this paper and the data will help decision-makers,  stakeholders, and the Ministry of Environment, Water, and Agriculture to protect the marine area.

  • You commented “Why is the letter M inserted in the Table 1? Whether it is necessary?”

The M was intended to represent the mean of the data.  This has been clarified in the table caption.

  • You commented “Another comment also applies to Table 1. The authors have provided appropriate references but adding the year of the study (in the table under/next to literature reference) of a particular metal would give more information.”

The years were added to table 1 many thanks for the very appropriate suggestion.

Reviewer 3 Report

The manuscript of Halawani et al., describe metal concentrations in metals and  Enrichment and Associated Risk Indices in sediments of  the Jeddah coast, eastern Red Sea near Jeddah, Saudi Arabia. The methodology is apprippriate as well as statistical analysis, I have some suggestions to be incorporate in the manuscript to improve it. 

1) Please indicate the novelty of the manuscript.

2) If possible consider to analyze sediments distribution through MDS (mustidimensional scaling).

Author Response

  • You commented “Please indicate the novelty of the manuscript.”

The authors appreciate this comment and agree the novelty and associated broader impacts could be more explicity stated.  As such we have added a section to the introduction and to the conclusion. Both sections follow here:

Coastal areas are considered sequestration points for various pollutants generated from commercial and urban activities. Different human-induced pollutants are typically delivered to coastal sediments through atmospheric transport and fluvial processes. A higher concentration of contaminants, including heavy metals, leads to deteriorating environmental parameters, including those gauging marine ecosystem health. The state of heavy metals and their distribution in coastal sediments, therefore, needs to be identified by applying comprehensive indices for categorizing the pollution level to inform appropriate future management decisions. The data and interpretation provided here will yield valuable benchmarks, not just for the immediate area, but for the region as a whole and to similar tropical marine ecosystems worldwide.

Heavy metals in near-shore marine sediments and their associated presence in marine waters are a threat to marine ecosystems.  A comprehensive evaluation of heavy metals contamination in such sediments, therefore,  is prudent for sensitive regions in close proximity to development. The results of this study demonstrate the use of different environmental indicators to identify current environmental health status and risk associated with anthropogenic activities. Further, bi- and mutlivariant statistical techniques suggest sourcing of coexpressed pollutants.  The findings will provide a valuable set of benchmarks for future research. Additionally, they will be a highly valuable guide to environmental decision makers in KSA and counterparts in other regional countries. Finally, the techniques employed and the correlations demonstrated can further aid similar work in analogous ecosystems worldwide.

  • If possible, consider to analyze sediments distribution through MDS (multidimensional scaling).

MDS analysis has been added as seen in Table 1 and Figure 4.  Associated section in the Methods, Results and Discussions section have also been added.
